# Howard Flack and the Flack Parameter

**David John Watkin * and Richard Ian Cooper** 

Chemical Crystallography Laboratory, Department of Chemistry, University of Oxford, Mansfield Road, Oxford OX1 3TA, UK; richard.cooper@chem.ox.ac.uk

*   Correspondence: david.watkin@chem.ox.ac.uk

**Abstract:** The Flack Parameter is now almost universally reported for all chiral materials characterized by X-ray crystallography. Its elegant simplicity was an inspired development by Howard Flack, and although the original algorithm for its computation has been strengthened by other workers, it remains an essential outcome for any crystallographic structure determination. As with any one-parameter metric, it needs to be interpreted in the context of its standard uncertainty.

**Keywords:** Howard Flack; Flack parameter; structure analysis; X-ray crystallography

## 1. Introduction

The formation of Howard Flack (1943–2017) was as a chemist, but he was also an able mathematician who turned this skill to crystallographic problems (Figure 1). Most of his contributions to crystallography will pass unnoticed by chemists relying on X-ray analysis to robustly characterise new materials and are probably not well known even to many professional structure analysts. He did not grind out an endless stream of papers, but instead produced a small number of carefully written, carefully thought out works in the domains of data acquisition, space group theory and crystallographic twinning.

Much of Flack's output has gone into the infrastructure of crystallography and need not concern structural chemists, but two of his insights have an immediate impact on ordinary structure analyses—correcting the observed data for absorption effects, and evaluating the absolute structure of chiral materials. An excellent technical review of the physics and mathematics behind absolute structure determination can be found in [1].

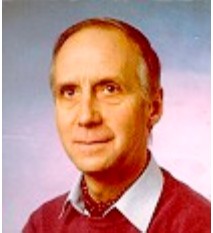

**Figure 1.** Howard Flack, courtesy of his widow Evelyne.

## 2. Impact on Service Crystallography

### 2.1. Background

Modern atomic-resolution small molecule X-ray structure analyses produce an enormous number of experimental observations even from relatively simple chemical substances. This wealth of data has enabled two procedures first described by Flack to become universally accepted as part of routine structure analyses.

Unlike specular reflection of light by a mirror, interaction between a crystal and X-rays is an interference phenomenon, which means that 'reflections' (actually diffracted beams) only occur when the crystal is orientated to satisfy Bragg's Law. Every diffracted beam (identified by the Miller Indices *h,k,l*) has a unique relationship with the unit cell of the crystal and an intensity dependent upon the internal make-up of the sample. Diffraction occurs at the Bragg Angles ($\theta$) given by:

$$\left(\frac{\sin\theta}{\lambda}\right)^2 = hG^*h^t \tag{1}$$

where G* is the reciprocal metric tensor (characteristic of the crystal) and h is the row vector of indices. *h,k,l* are positive and negative integers permutated up until the maximum $\sin\theta = 1$. In practice there are experimental limitations below this maximum, but even so, X-ray crystal structure analysis is *data rich*, that is to say, there are very many more observations available than there are parameters describing the structure. For much of the 1960s, 70s and 80s most X-ray data were measured on instruments which observed reflections one at a time (serial diffractometers). Depending upon the space group of the sample, some (and perhaps many) of the permutated indices refer to *equivalent reflections*. Equivalent reflections are ones for which the theoretical intensity should be identical even though the indices are different. Thus, in the space group *Pc* the reflection *h,k,l* will be equivalent to *h,−k,l*. Because of the time taken to measure data, it was common practice to only measure one of a group of equivalent reflections, thus, in the example above, halving the data collection time. When more than one equivalent reflections are measured, after correction for experimental geometry and short comings, they are averaged (merged) by crystallographic analysis software to provide a set of *unique* observed reflections. $R_{merge}$ and $R_{int}$ are measures of the self-consistency of groups of equivalent reflections. Current area detector diffractometers measure hundreds of reflections quasi-simultaneously, so that many equivalent reflections can be observed.

In a structure determination, the analyst postulates an atomic model (usually via a *structure solution* program), generally with 9 parameters per independent atom. A structure factor can be calculated from this model, and the parameters optimized to match $I_c$ with $I_o$ (see Appendix A) by the method of nonlinear least squares. In a modern analysis, the observation: parameter ratio is often in the range 10 to 20, equivalent to fitting a straight line ($y = mx + c$) to between 20 to 40 observations.

Just as the image of an object remains visible when a mirror is rotated about its normal, so diffraction continues to occur as the crystal is rotated about the normal to the diffracting plane. This means that on a suitably engineered instrument, (e.g., a four circle diffractometer), the rotational orientation ($\psi$) of the crystal can be varied for a given Miller index. Before the widespread use of area detectors, about 25 years ago, only a single measurement was generally made for each unique Miller index. Modern instruments are normally programmed to make between 5 and 20 observations of equivalent reflection at different $\psi$ values. These independent measurement of the same quantity are termed redundancy, or multiplicity of observation, MoO. Measurements on a typical organometallic crystal might take a less than an hour on a laboratory instrument, or just a few minutes at a synchrotron source, yielding a data set of tens of thousands of observations.

## 2.2. Correction for Absorption

One problem facing all X-ray structure analysts is 'correcting' their observed data for experimental effects. Some of these, such as the Lorentz and polarisation corrections, have been well characterised analytically since the 1920's. More problematic has been computing corrections for absorption effects. X-rays are attenuated as they pass through a medium according to Beer's Law:

$$I_o = I_i e^{-\mu t}, \tag{2}$$

where $I_o$ is the observed X-ray intensity, $I_i$ is the incident intensity, $\mu$ is the absorption coefficient (computed from the elemental composition [2]) and *t* is the path length through the sample. Unless the

crystal can be ground to a sphere (which is still sometimes done for special purposes), the exponential means that exact mathematical calculations depend upon very accurate measurements of the crystal shape and size—still not easily done even with modern digital microscopes.

For an X-ray beam travelling along the long axis of a prismatic crystal, the intensity of the emergent beam will reduce and then increase again as the crystals is rotated through an angle ψ about a vector perpendicular to the long axis. In the 1960s, methods based on experimental observations were developed for making empirical corrections for absorption by tracking the intensity of a chosen reflection as ψ was systematically varied [3,4]. These methods created a kind of calibration curve for the sample. Flack recognised that these curves could be better represented by a smooth mathematical function [5]. Because of the periodic nature of the curves, one natural function of choice was the Fourier series. Flack's careful experimental verification of the method, which included making thousands of observations on a serial diffractometer, demonstrated the strength of the concept. Today, area detector diffractometers inevitably measure many reflections at several different values of ψ. While these differing ψ values almost never correspond to systematic curve tracking, their great abundance enables them to be used is a similar manner. Blessing replaced the Fourier series by spherical harmonics [6], and this has become the basis of all experimental data corrections up to modern times. It turns out that this empirical method of correcting for absorption can also be used to correct for a range of other experimental problems, and since modern area-detector diffractometers provide a large number of equivalent observations (high MoO), the corrections are generally applied without any special action from the analyst. The robust correction of the observations is an important contributor to a robust determination of the absolute structure of a crystal.

### 2.3. Determination of Absolute Structure

However, for most chemists working in chiral chemistry, it is Flack's parameter that will most evidently impact them. It had been known since the 1930's that under specific conditions the diffraction of X-rays from a non-centrosymmetric crystal displays small effects which reflect the symmetry of the crystal, and hence the symmetry of the materials the crystal was built from [7]. Of special interest to the chemist working with chiral materials is the fact that for crystals of these materials, the diffracted intensities of a pair of reflections $h,k,l$ and $-h,-k,-l$ (which are identical for crystals in centrosymmetric space groups) are subtly different [8]. The difference between the members of these Friedel pairs are generally very small and so easily masked by experimental issues. The magnitude of these Friedel differences depends upon the resonant scattering of the elements in the material and the wavelength of the X-rays. Early attempts to determine the absolute structure of a material depended upon the analyst identifying enantiomer sensitive pairs and then carefully remeasuring them. Hamilton's 'R-factor Ratio Test' tried to use all the observable reflections [9], but was difficult to apply reliably. A breakthrough was Roger's proposal that the absolute structure of a material could be estimated, together with a reliability index, by the determination of one additional parameter (which he called η) in the model [10]. Unfortunately, the derivative of this parameter with respect to the calculated structure factor is discontinuous over the physical range of the parameter, meaning that its interpretation was unsound except for truly enantio-pure materials. Flack became interested in this mathematical problem, and within two years had reformulated it in terms of a new parameter (which he called '$x$', see Section 4 for derivation) which had a continuous derivative, and thus could be used in the analysis of crystals which, while containing only chirality-preserving symmetry operations, contained a mixture of domains of two enantiomers—in crystallography called '*twinning by inversion*' [11]. When $x$ is zero, the material is enantiopure and the atomic coordinates determined from the data correspond to the actual structure. If $x$ is unity, then the atomic model is the inverse of the actual structure—the model is 'inverted'.

### 3. Twinning by Inversion

It has been estimated that when solutions contain exactly equal amounts of both enantiomers of a compound form crystals, approximately 90% will form racemic crystals, containing exactly equal numbers of each enantiomer [12]. These may be related by a centre of inversion—the centrosymmetric crystals drawn from the set labelled CA in Figure 2, or by chirality-inverting non-centrosymmetric operations in non-centrosymmetric space groups (crystal classes labelled NA in Figure 2) for example space group *Pc*.

Occasionally, as Pasteur had the good fortune to observe, the two enantiomers preferentially separate out into enantiopure crystals, a phenomenon called spontaneous resolution. The enantiopure constituent molecules restrict the choice of space group to one of the 65 Sohncke groups which contain exclusively chirality-preserving symmetry operations, i.e., rotations and translations (from the crystal classes labelled NC in Figure 2).

In rare cases both enantiomers may crystallize together in a Sohncke space group as independent molecules which are not related by a symmetry operation of the crystal. Approximately 50 examples of this type were identified in Flack's review of chiral and achiral crystal structures in 2003 [13]. Materials exhibiting this phenomenon have subsequently been termed kryptoracemates, as their racemic nature is hidden by the space group symmetry. A recent survey identified 724 examples in the crystallographic literature [14].

In addition to these cases, it is also possible to form a solid solution of enantiomers. The crystal grows without chiral selectivity so that either enantiomer can occupy a molecular site within a crystal structure resulting in a partially disordered solid solution of enantiomers [15].

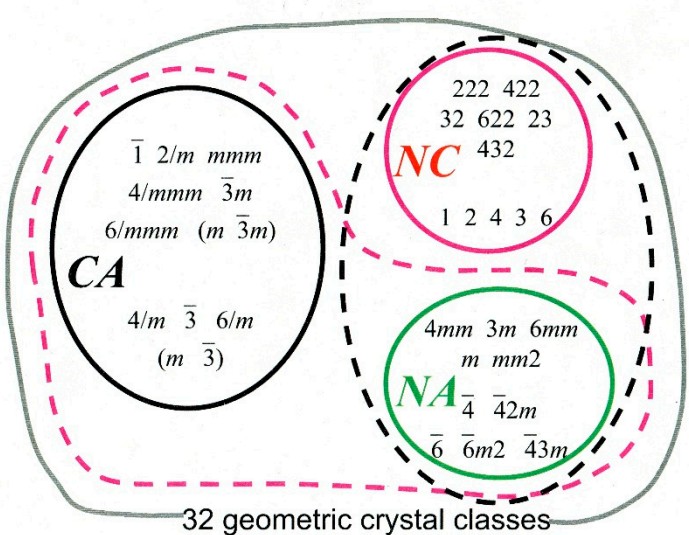

**Figure 2.** The 32 geometric crystal classes. CA means Centrosymmetric & Achiral, NA means Non-centrosymmetric & Achiral, NC means Non-centrosymmetric & chiral. Note that the symbols in the achiral classes contain either an m, $\bar{4}$ or $\bar{6}$ operator, allowing pairs of enantiomers to be related by a chirality-inverting symmetry operation which is not a centre of symmetry. Figure by courtesy of Evelyne Flack [16].

If the solution contains one and only one enantiomer, it will crystallise in a Sohncke space group. In this case, the determination of Flack's parameter will enable the stereochemistry of the material to be estimated. Note well that the stereochemistry is only estimated. Refinement, by least squares, of all the parameters needed to characterise the material (the positions of the atoms in the unit cell, their motion about their mean positions, and Flack's parameter) provides both values for these parameters and estimates of their standard uncertainties (written u or s.u.), previously called standard deviations. This is really important. For example, the standard uncertainties in atomic positions enable one to

compute the standard uncertainty in the distance between them, which in its turn enables a rational decision to be made as to whether two bond lengths are significantly different. Flack's parameter is also determined with a standard uncertainty, enabling the analyst to assign a level of confidence to the proposed absolute structure.

It is not clear when 'Flack's x parameter' became simply 'The Flack parameter'. For the 20 years following its formulation, the determination of the Flack parameter required extreme attention to experimental details. For first row elements the magnitude of the resonant scattering is small for copper radiation ($\lambda$ = 1.54 Å) and even smaller for molybdenum radiation ($\lambda$ = 0.71 Å), and so it was generally held that the absolute structure of light atom materials required copper radiation. It was not uncommon at that time to form a derivative by inserting an atom with larger resonant scattering (such as chlorine) into the molecule. One snag with this approach was that heavier atoms tend to have higher absorption coefficients—though the absorption corrections described above tend to minimize the impact on the data quality. The millennium saw the widespread use of area detectors for the measurement of diffraction data at temperatures close to the boiling point of liquid nitrogen. Working at this temperature reduces the motion of the atoms in the molecules, and greatly improves the quality of the experimental observations. These instruments measure the intensities of very many Bragg reflections quasi-simultaneously rather than measuring the reflections one-by-one. As well as providing the many observations needed for the absorption corrections, these instruments also measured many of the Bijvoet (commonly used as a synonym for Friedel) pairs of reflections needed for the determination of Flack's parameter (see Appendix B). The determination of absolute structure by the determination of the Flack Parameter became common place.

If the enantiopurity of the sample is unquestionably known by other experimental methods, then an estimate of the Flack parameter by any diffraction calculation is likely to be sound. Detailed evaluation of modern data from light atoms structures of known chirality has shown that even analyses using molybdenum radiation are unlikely to point to an incorrect absolute structure [17].

Problems arise if the enantiopurity is uncertain. Unless instructed otherwise, most X-ray analysts search through the sample for the 'best' looking crystal, i.e., one with bright faces, roughly isometric shape, no inclusions, etc. The outcome of the analysis is the absolute structure of that particular crystal. Nothing can be said about the nature of the other crystals in the batch, nor of the solution from which the crystals were grown. Other, non-crystallographic, methods must be used to verify that the absolute structure of the sample crystal is indeed the same as that of the whole batch. Although the crystal used in the X-ray analysis may measure less than 0.1 × 0.1 × 0.1 mm, it is now possible to verify by NMR, chromatography or optical methods that the single crystallographic specimen really represents the bulk sample [18]. Without a references sample or sophisticated simulations such methods cannot directly determine the absolute configuration of enantiomers, however these methods are extremely valuable for determining the enantiomeric excess of samples which can be used as a valuable piece of prior information in a crystallographic absolute structure determination.

Thompson et al. tabulated all the possible outcomes from samples of unknown purity [19]. The bottom line is that for a sample which is not enantiopure, these cases may occur singly or together:

(1) Spontaneous resolution, with separate crystals forming of each enantiomer.
(2) Pairs of enantiomers may crystallize together form racemic crystals, with any excess concentration of one enantiomer forming enantiopure chiral crystals.
(3) Formation of inversion twinned crystals, which contain contiguous domains of sufficient size to diffract coherently, some of which are related by inversion.
(4) The material is achiral but forms chiral crystals.

The crystallographer can do little about cases 1 and 2 except pay attention to the morphology of the crystals in the sample. In case 1 crystals may grow, as they did for Pasteur, having small facets which are mirror images in the two enantiomeric crystals. In case 2 it is very likely that the achiral crystals will look quite different to the enantiopure chiral crystals. It is now widely accepted that case 3

is uncommon for organic materials, but may be an issue for inorganic or organo-metallic materials where chirality of the sample may arise from the arrangement of the building blocks in the crystal [20]. Where twinning by inversion does exist, it may be clearly identifiable under an optical microscope, or it may be on a sub-microscopic scale with small zones of each enantiomer distributed throughout the crystal. Case 4 is illustrated by silica. The building block of quartz, $SiO_4$, is achiral but these motifs can be assembled in crystals to form left or right handed spirals. The hand of the spiral sometimes reveals itself as small facets on the surface of well-formed crystals but is extremely rare [21].

## 4. Interpretation of the Flack Parameter

The equation defining the Flack Parameter [11] is simply:

$$I_o \cong I_c = (1-x)I_s^+ + xI_s^-, \tag{3}$$

where $x$ is a kind of partition or mixing coefficient with extreme values of zero or unity. $I_o$ is the observed structure amplitude, $I_s$ is the amplitude computed for a single enantiomer, $I_c$ is the amplitude computed for the twin. The crystallographer develops an atomic model representing the structure (i.e., one which can be displayed in a graphics program) and from this can compute structure amplitudes ($I_s^+$ and $I_s^-$) for the structure and its inverse.

$x$ can be included as a parameter in the structure refinement, or Equation (3) can be solved for $x$ by least squares from all the reflections once $I_c^{\pm}$ are known. The first method is that used by Flack in his original work [11]. Sheldrick developed a post-refinement method for estimating the parameter as a one-off computation once the main refinement was completed which he called the "Hole in One" method [22]:

$$I_o^+ - I_s^+ \cong x\left(I_s^- - I_s^+\right) \tag{4}$$

Later Hooft et al. [23] re-wrote the expression to use the differences of Friedel pairs and developed a Bayesian statistical post-refinement method:

$$D_o \cong (1-2x)D_s, \tag{5}$$

where $D_o = I_o^+ - I_o^-$ and similarly for $D_s$. Parsons included the averages of the observed and of the computed Friedel pairs as a way of reducing the influence of experimental errors—the method now widely known as Parsons' Quotients in Equation (6) [24]:

$$Q_o \cong (1-2x)Q_s, \tag{6}$$

where $Q_o = \left(I_0^+ - I_0^-\right)/\frac{1}{2}\left(I_0^+ + I_0^-\right)$ and similarly for $Q_s$.

For a while these post-refinement methods were treated with suspicion because they lacked the estimates of covariance between the atomic parameters and Flack's parameter which classical full matrix refinement provided. Leverage analysis of the full matrix for refinement of the Flack parameter together with the atomic parameters identifies those reflections most influential in determining $x$ [25]. Ironically, it turns out that relatively few reflections carry strong information about the Flack parameter—very careful remeasurement of these reflections should reduce the standard uncertainty of $x$. Eventually, an in-depth analysis by Parsons et al. showed that except in marginal cases the covariance was small, and that post refinement methods were generally safe to use [26].

Most crystallographic programs enable you to compute (with greater or lesser ease) the Flack parameter by both direct refinement in the main analysis (Flack's original method), and by one or more of the post refinement methods. Generally, the methods give substantially the same result, except the standard uncertainty of the direct refinement can be about twice as large as post refinement methods. Where there is a significant difference either in the value of $x$ itself or in the standard uncertainty, this can sometimes be traced back to problems with the analysis [27]. These problems are masked in the main refinement because the observations are given weights which include both a contribution from

the observed variance of the reflection and a modifier chosen so that the weighted residual, $w\left(F_o^2 - F_c^2\right)^2$, is fairly uniform across all the data. This modifier is justified in that it accounts for unidentified short comings in both the data and the model, and yields a linear normal probability plot. Post refinement methods generally give a linear normal probability plot using the unmodified variances.

## 5. Conclusions

*Practical Advice Concerning the Flack Parameter*

It is crucial to make it clear to the X-ray analyst from the outset that the absolute structure needs to be determined. Although modern instruments routinely yield exceptionally high-quality data, data collection parameters can be further optimized for the determination of absolute structures. Since the X-ray analysis will usually be made on just one single crystal selected from the material submitted, it is important to provide an estimate the sample's enantiopurity. This will warn the analyst about the need to look out for crystals with an anomalous morphology. Where possible, recover the actual crystal used in the X-ray analysis and verify that it represents the bulk sample. It is no longer fashionable to make heavy atom derivatives since it is preferable to avoid unnecessary additional synthetic processes. However, it does not seem to be widely appreciated that the heavy atom does not have to be part of the material under investigation. Samples containing solvent of crystallization where the solvent contains a heavy atom (e.g., $CH_2Cl_2$) will generate larger Bijvoet differences.

The determination of a parameter from many thousands of observations by the method of least squares means that as well as getting an estimate of the parameter value, one also gets an estimate of its reliability—the standard uncertainty, e.g., $0.05 \pm 0.01$, written by crystallographers as 0.05(1).

The standard uncertainty is key to interpreting the value of *x*. If the parameter is determined with a large s.u. then the value of *x* is unreliable and caution should be used when drawing conclusions about the enantiopurity of the sample. Flack and Bernardinelli argued that for a material known to be enantiopure, a standard uncertainty of less than 0.08 is sufficient for the assignment of absolute structure to be well determined [28]. A chirally sensitive HPLC or NMR experiment can establish the ratio of enantiomers in a sample, but not the absolute configuration of the dominant enantiomer. If the enantiopurity is unknown at the time of the crystallographic analysis, the s.u. needs to be below 0.04 to indicate a confident outcome. Even here, there is room for uncertainty. If the refinement gives $x = 0.10(4)$, this can be interpreted as *x* lying within 3 s.u. of zero, so the absolute configuration is confirmed. However, it can also be interpreted as meaning that the sample contains up to 10% of the opposite enantiomer in the form of a racemic twin. X-ray crystallography can provide a robust indication of the absolute structure of the bulk of the single crystal used in the experiment, not the absolute structure of the bulk of a batch crystalline sample, and it can only provide an estimate of the enantiopurity (with standard uncertainty).

**Author Contributions:** Conceptualization and production, R.I.C. and D.J.W. All authors have read and agreed to the published version of the manuscript.

**Funding:** This research received no external funding.

**Conflicts of Interest:** The authors declare no conflict of interest.

## Appendix A. Observed and Calculated Structure Amplitudes

If an atomic model exists for a crystalline material, a calculated structure factor can be computed from:

$$F = A + iB$$

and hence:

$$F^2 = A^2 + B^2$$

Note that the structure factor is a complex quantity, having both a magnitude and a phase. The squared term is the calculated structure intensity which is proportional to the observed structure intensity. The magnitude can easily be derived from the intensity, but in general the phases are unknown.

In the older literature, the observed intensity usually referred to the physically measured magnitude squared of a given reflection and was represented by the symbol $I_o$. This was converted to an observed squared structure factor by the application of various experiment-dependent correction factors, represented by the symbol $F_o^2$, and the corresponding term computed from the atomic model was represented by $F_c^2$.

Increasingly (and confusingly), following the usage in the macromolecular community, $I_o$ is used to represent $F_o^2$, and similarly for $I_c$. We have followed this convention in this manuscript, with the additional extension that $I_s$ represents the squared structure amplitude computed for an enantiopure material, and $I_c$ allows for twinning. See Equation (3).

**Appendix B. Friedel and Bijvoet Pairs**

A definition gaining traction is that Friedel pairs are strictly related by inversion (e.g., *h,k,l* and *−h,−k,−l,*), where as a Bijvoet pair is any symmetry equivalent related by a symmetry operator of the second kind (e.g., *h,k,l* and *h,−k,l*) [29].

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
