# Peer review of "Howard Flack and the Flack Parameter"

_chemistry, doi:10.3390/chemistry2040052_

Round 1

Reviewer 1 Report

Watkin and Cooper's review article is an excellent contribution to the special isssue dedicated to the memory of Howard Flack. I enjoyed reading it. I recommend publication after minor revisions, as indicated below.

Title: Since the scope of the article is actually wider than just the Flack parameter, one could alter the title. Maybe "Howard Flack and absolute structure determination" or something like that. Just a suggestion.

Abstract: "It's" should be "Its" (line 9).

Main text:

Page 1, line 16: the figure is never referred to in the text. I also recommend placing it after the first paragraph of the introduction. Moreover, "Figure 1" is also used for the figure on page 3. So the count numbers of figures need to be adjusted.

Page 1, line 23: "twining" should probably be "twinning".

Page 1, line 33: the meaning of "high resolution" is not clear. In the context of routine X-ray crystallography it would probably mean "atomic resolution" rather than actually high resulotion X-ray diffraction data as in charge density studies. I think that "(High Resolution)" can be left out altogether here.

Page 2, line 40: maybe "h,k & l" should be "h, k and l" or just "k,k,l".

Page 2, line 56: for novices or readers without background in X-ray crystallography, the half sentence regarding merging (averaging) of reflections could be somewhat misleading. It should be made clear that the data are only merged to calculate Rint etc., whereas unmerged datasets should be used for routine refinement and data deposition.

Page 2, line 61: I am aware that the border between traditional direct methods and new dual-space methods is not clear-cut, but I think that "usually via a direct methods" does  not hold anymore. My impression is that 90 % of small molecule structures are solved with SHELXT nowadays. Again, I would leave out "(usually via a Direct Methods program)" altogether.

Page 2, equation 2: At a first glance, Beer's Law looked strange to me, because I had mistaken Io for I0. I recommend writing it in the same way like in most textbooks or the International Tables: I = I0e–μt, where I0 is the intensity of the incident beam.

Page 2, line 85: since the calculation of the linear absorption coefficient, which also depends on the wavelength of the X-ray, is black-boxed in programs like SHELXL or PLATON/checkCIF, it would be nice to add at least a reference where the interested reader could look up the details of the calculation. Maybe the respective chapter of the International Tables?

Page 3, figure 1 (or actually fig. 2, see above):

  • The figure caption (i.e. lines 90, 91 and 92) should be properly set and I disagree with "redrawn from Caspari". That needs clarification. Ref. 2 does not contain the figure as is. Obviously, just the crystal shape was redrawn from the Caspari 1926 article. "Alpha-quinol" is not clear to me. In the paper it is just "quinol", which would be hydroquinone or benzene-1,4-diol (IUPAC) nowadays. That should be clarified and preferably a IUPAC name of the compound should be given (at least in parentheses).
  • It is not clear to me whether the line in the figure representing the intensity of the X-ray beam is just schematic or is actually based on the data in the Caspari article. Maybe axes could be added: ordinate with intensity (a.u.) and abscissa with orientation (°), I guess.

Page 3, line 99: in general, the readability would be improved if the citation were placed at the end of the sentence - if possible. This holds for the entire manuscript.

Page 3, line 106: no need to capitalise "spherical harmonics" here.

Page 4, line 132: remove the hyphen from "enantiopure".

Page 4, line 134: maybe equation 3 should be placed on this page, as it is referred to here for the first time?

Page 4, line 136: is the half-sentence "in crystallography called 'twinning by inversion'" necessary here? I would make sense to me if just cyrstals with Sohncke space groups are meant, because I think that non-centrosymetric achiral space groups would contain a 1:1 ratio of both enantiomers anyway.

Page 4, line 142: I think that "Normally" here actually means that an estimate of 90 % of racemates form crystals containing both enantiomers and 10 % conglomerates (e.g. https://dx.doi.org/10.1007/978-94-011-0353-4_12 ). So "normally" should be defined a little more precisely here, maybe with some references.

Page 4, line 149: I recommend also mentioning the term "kryptoracemate" here and, aside from ref. [12], the authors should not neglect the very recent study on the topic by Toms Rekis: http://scripts.iucr.org/cgi-bin/paper?S2052520620003601

Page 4, line 154: I think the bar over the number in e.g. -4 does not indicate moieties related by a centre of symmetry and I do not understand "a lower case letter". I think that m rather than a lower case letter represents a mirror plane. Maybe italic letter rather than lower case letter was meant?

Page 4, line 157: I know it seems like nitpicking but some crystallographer colleagues and structural chemists dislike "chiral space group" as synonym for Sohncke space group and they are right - as you know. See: https://dictionary.iucr.org/Chiral_space_group So would replace it by "Sohncke space group" here.

Page 5, line 168: The sentence "It is not clear when 'Flack's x parameter became simply "The Flack parameter" seems somewhat disconnected here. It should be removed or placed elsewhere in the text.

Page 5, lines 172 and 173: a space before Angström is probably missing.

Page 5, line 200: "NMR" should be capitalised. I guess that optical methods like polarimetry or CD spectroscopy are only of help for that purpose if enantiopure reference samples are available, which is usually not the case for newly synthesised compounds. Maybe the investigation of the enantiopurity of the bulk sample (ee value etc.) deserves a few sentences of explanation here.

Page 5, line 211: Maybe the term "kryptoracemates" could also be added here at least in parentheses. A fifth item in the list could be a solid solution of enantiomers.

Page 5, line 216: It was interesting to learn that it is now widely accepted that twinning by inversion is uncommon for organic materials. It seems to agree with my own experience, but I have to admit that have not paid attention to the phenomenon yet. Is there a reference which could be added to support the statement?

Page 6, line 246: Could you perhaps also give an equation for Parsons's method?

Page 6, line 259: I think that the standard uncertainty of the Flack parameter determined by post-refinement methods should be about half of that of the refined value ( https://dx.doi.org/10.1107/S2052520616014773 ).

Page 7, line 268: "Conclusion" should perhaps be "Conclusions".

Page 7, line 298: an estimate of the enantiopurity of the crystal studied rather than an estimate of of the enantiopurity the bulk sample let alone the solution it crystallised from. That should perhaps be emphasised or clarified here.

I wonder wether the Appendices are needed. Maybe their content could be incorporated in the main text. Just a suggestion.

Appendix B: I have always considered Friedel pairs and Bijvoet pairs to be synonyms. That appears to conform to the IUCr dictionary of crystallography: https://dictionary.iucr.org/Friedel_pair Interesting that ref. 24 suggests otherwise?

Page 8, line 336: The complete title of the article is: "CCCXCIII. The Crystal Structure of Quinol. Part I.".

Page 9, line 375: ref. 17 should be replaced by George Sheldrick's 2008 Acta Cryst. A article entitled "A short history of SHELX": https://doi.org/10.1107/S0108767307043930

Page 9, line 388: The title is missing.

Author Response

We thank the referee for the comments. Our responses are inserted
below - Please see the attachment. An updated manuscript has been uploaded.

Reviewer 2 Report

The publication 'Howard Flack and The Flack Parameter' is a study on the problems / issues that a crystallographer must face in his work. Such a study can be very valuable especially for young researchers who prepare their diploma theses. The structure of the publication is well planned and divided into sections for easy reading, the work seems interesting and deserves to be published. However, I have some comments that should be considered by the authors.

  1. The editorial side needs to be refined - I didn't notice any errors or typos, but double spaces are common.
  2. Mental shortcuts are used that do not make it easier to read the work.
  3. line 74-76: Measurements on a typical organometallic crystal might take less than one hour on a laboratory instrument - pleas consider.
  4. line 91-92: Is the text part of the drawing? Or signature under the drawing? This part is very confusing.
  5. The space group Pc is always selected as an example. Is there any practical reason? (I ask out of curiosity)
  6. The part titled '3. Twinning by Inversion' raises doubts. It is not clear to me what definition the authors use. Can I ask for an explanation?
  7. line 142-150: if solutions contain exactly equal amounts of both enantiomers of a compound we can receive (1) racemic crystals, or (2) mixture of enantiomeric crystals, or (3) mixed crystals (solid solutions of enantiomers) containing non-stoichiometric amounts of enantiomers in the crystal.
    Indeed, racemic crystals can be divided into centrosymmetric, non-centrosymmetric and crystallizing in chiral groups - according to the literature they are called kryptoracemates. I disagree that there are only 50 examples of this type of crystals. As reported in Acta Cryst. (2010). B66, 94–103, there are more of them, and it is possible that 10 years later the list is even longer. It seems to me that the literature should be updated.
  8. line 209-210: is it just solid solutions of enantiomers? (the definitions used on lines 206-211 are not clear).
  9. It seems unlikely to me to observe an twinning by inversion under a microscope. Can I ask you to clarify this issue?
  10.  Regarding the conclusions: unless we are sure of the highest quality of measurement, if we want to confirm a slight admixture of the second enantiomer in crystals, we should always seek support from other analytical techniques - for example HPLC. Besides, from a statistical point of view, the parameter x=0.10(4) is irrelevant.

Author Response

(The authors gave the same response as above.)

Round 2

Reviewer 2 Report

Thank you for considering my comments. Unfortunately, the work is still full of mental shortcuts.
I still have doubts about the observation of the inversion twin under the microscope. Of course, I agree that twinning can (and often is) visible, especially in polarized light. However, as long as we do not perform a diffraction measurement, we cannot be sure that it is the twinning by inversion. It seems to me that this should be highlighted. Especially since the authors themselves claim that the publication was written for non-specialists.

I think there is a typo on line 146 - shouldn't there be "the model is inverted.”?

In line 170 it should read: "partially disordered solid solution of enantiomers”.